# Identification of Novel Pathways Associated with Patterned Cerebellar Purkinje Neuron Degeneration in Niemann-Pick Disease, Type C1

**DOI:** 10.3390/ijms21010292

**Published:** 2019-12-31

**Authors:** Kyle B. Martin, Ian M. Williams, Celine V. Cluzeau, Antony Cougnoux, Ryan K. Dale, James R. Iben, Niamh X. Cawley, Christopher A. Wassif, Forbes D. Porter

**Affiliations:** 1Section on Molecular Dysmorphology, Division of Translational Medicine, Eunice Kennedy Shriver National Institute of Child Health and Human Development, National Institutes of Health, Department of Health and Human Services, Bethesda, MD 20892, USA; ky.martn@gmail.com (K.B.M.); ian.williams0108@gmail.com (I.M.W.); celine.cluzeau79@gmail.com (C.V.C.); antony.cougnoux@nih.gov (A.C.); cawleyn@mail.nih.gov (N.X.C.); wassifc@cc1.nichd.nih.gov (C.A.W.); 2Bioinformatics and Scientific Programming Core, Eunice Kennedy Shriver National Institute of Child Health and Human Development, National Institutes of Health, Department of Health and Human Services, Bethesda, MD 20892, USA; ryan.dale@nih.gov; 3Molecular Genomics Core, Eunice Kennedy Shriver National Institute of Child Health and Human Development, National Institutes of Health, Bethesda, MD 20892, USA; james.iben@nih.gov

**Keywords:** Niemann–Pick disease, type C1, NPC1, cerebellum, neurodegeneration, Purkinje neurons, RNA-seq, WGCNA

## Abstract

Niemann-Pick disease, type C1 (NPC1) is a lysosomal disease characterized by progressive cerebellar ataxia. In NPC1, a defect in cholesterol transport leads to endolysosomal storage of cholesterol and decreased cholesterol bioavailability. Purkinje neurons are sensitive to the loss of NPC1 function. However, degeneration of Purkinje neurons is not uniform. They are typically lost in an anterior-to-posterior gradient with neurons in lobule X being resistant to neurodegeneration. To gain mechanistic insight into factors that protect or potentiate Purkinje neuron loss, we compared RNA expression in cerebellar lobules III, VI, and X from control and mutant mice. An unexpected finding was that the gene expression differences between lobules III/VI and X were more pronounced than those observed between mutant and control mice. Functional analysis of genes with anterior to posterior gene expression differences revealed an enrichment of genes related to neuronal cell survival within the posterior cerebellum. This finding is consistent with the observation, in multiple diseases, that posterior Purkinje neurons are, in general, resistant to neurodegeneration. To our knowledge, this is the first study to evaluate anterior to posterior transcriptome-wide changes in gene expression in the cerebellum. Our data can be used to not only explore potential pathological mechanisms in NPC1, but also to further understand cerebellar biology.

## 1. Introduction

Niemann–Pick disease, type C (NPC) is a rare autosomal recessive disorder characterized by accumulation of unesterified cholesterol and glycosphingolipids within the endo-lysosomal system. Although the age of onset and clinical manifestations are heterogenous, NPC patients may present with neonatal cholestasis or hepatosplenomegaly. These visceral signs often precede the onset of progressive neurological disorders, which include vertical supranuclear gaze palsy, dementia, and cerebellar ataxia [1,2]. NPC is caused by mutations in either *NPC1* or *NPC2*, which encode a lysosomal transmembrane protein and a small soluble lysosomal protein, respectively [2,3]. Together, the NPC1 and NPC2 proteins function to facilitate the efflux of cholesterol from the late endosome-lysosome compartment, as mutations in either gene leads to the accumulation of unesterified cholesterol, glycosphingolipids, and additional lipids within this compartment [2,4,5]. However, the pathological mechanisms by which these mutations lead to progressive neurodegeneration remains unclear.

A common neurological complication observed in NPC1 patients is cerebellar ataxia [6]. The cerebellum is a highly organized and conserved structure that, in mammals and birds, is divided into ten distinct lobules by shallow fissures. Although each lobule has a distinct form, each is comprised of three layers. The center layer is the Purkinje neuron layer, which derives its name from the monolayer of Purkinje soma that aligns within each lobule. The Purkinje neurons extend elaborate dendrites into the outermost molecular layer, where they form excitatory synapses with climbing fibers that originate from the inferior olivary nucleus. The innermost granule cell layer is mainly comprised of granule cells, which are a densely packed neuronal cell type. Granule cells account for the largest neuronal population within the brain. In the cerebellum, granule cells function to transmit excitatory input from mossy fibers to the dendrites of the Purkinje neurons through their long axonal projections, known as parallel fibers. Hence, synaptic inputs from granule cells and the climbing fibers are integrated by the Purkinje neurons before being projected down their axons, where they make inhibitory synaptic contact with cerebellar nuclei at the base of the cerebellum. Therefore, Purkinje neurons represent the sole output of the cerebellum, as they integrate all cerebellar inputs before projecting to their target neurons [7].

Although the structure of the cerebellar cortex is consistent throughout all lobules, distinct patterns of gene expression generate highly reproducible transverse zones within the cerebellum that can further be divided into parasagittal stripes [8,9]. The most common marker of these zones and stripes is zebrin II, which has been identified as the glycolytic enzyme, aldolase C [8,10]. Zebrin II is a Purkinje neuron specific marker that is expressed in a distinct subset of aligned Purkinje neurons, thus resulting in an alternating array of zebrin II-positive and zebrin II-negative parasagittal stripes [11]. Furthermore, these stripes form larger domains that comprise four separate transverse zones of the cerebellum: the striped anterior zone (lobules I–V), the homogeneously zebrin II-positive central zone (lobules VI–VII), the striped posterior zone (lobules VIII–dorsal IX), and the uniformly zebrin II-positive nodular zone (ventral lobule IX–X) [9,12]. These unique gene expression patterns also give rise to functional differences, as many cerebellar disorders resulting in Purkinje neuron death are restricted to specific zones or parasagittal stripes [13]. Purkinje neurons within the zebrin II-negative stripes are more susceptible to cell death, resulting in an anterior-to-posterior gradient of neurodegeneration. This pattern is observed in multiple spontaneous mouse models, such as the *Purkinje cell degeneration* (*Nna1*^pcd^) [14], *tottering* (*Cacna1a*^tg^) [15], and *lurcher* (*Grid2*^lc^) [16] mice, or in response to numerous exogenous toxins, such as ibogaine [17], cytosine arabinoside [18], methotrexate [19], and alcohol [20,21,22]. Similar anterior-to-posterior Purkinje neuron death patterns are observed in multiple genetic disorders, including those for multiple forms of the spinocerebellar ataxias [23,24,25], Menkes syndrome [26,27], Niemann–Pick disease type A/B [28], and Niemann–Pick disease type C1 [29]. For NPC1, studies utilizing BALB/cNctr-*Npc1^m1N^*/J (*Npc1^−/−^*) mice have demonstrated that the Purkinje neurons undergo a patterned cell death that initially affects the zebrin II-negative Purkinje neurons starting at approximately postnatal day 40 [30]. As the disease progresses, additional zebrin II-positive Purkinje neurons throughout lobules I–IX are also lost. This progression continues until the animal succumbs to the disease, usually around postnatal day 70 to 90, at which time the only Purkinje neurons that remain are those zebrin II-positive cells of the ventral portion of lobule IX and the entire lobule X [31]. This spatiotemporal patterned cell death occurs despite the consistent presence of unesterified cholesterol accumulation and neuroinflammation throughout all the cerebellar lobules [32,33].

To gain mechanistic insight into factors that would either protect Purkinje neurons or potentiate their degeneration, we evaluated spatial cerebellar RNA expression patterns. Cerebellar lobules III, VI, and X were individually microdissected from asymptomatic 4.5-week-old *Npc1^+/+^* and *Npc1^−/−^* mice. RNA-sequencing was performed and weighted gene co-expression network analysis (WGCNA) [34] was utilized to observe any intrinsic, or NPC1-specific, differences in gene expression that may contribute to the survival of Purkinje neurons in lobule X of *Npc1^−/−^* mice.

## 2. Results

### 2.1. Immunohistochemistry and RNA-Sequencing

Immunohistochemistry of calbindin staining in the cerebellum of *Npc1^−/−^* mice confirmed the temporal and lobule specific loss of Purkinje neurons with disease progression when compared to *Npc1^+/+^* littermates, which have a consistent Purkinje neuron density throughout their lifespan (Figure 1A). At 4.5 weeks of age, the staining pattern of *Npc1^−/−^* Purkinje neurons demonstrated no signs of neurodegeneration, consistent with a pre-symptomatic state of the *Npc1^−/−^* mice (Figure 1B). At seven weeks of age, however, the Purkinje neurons of the anterior cerebellar transverse zone (lobules I–V) have mostly degenerated, leaving only sporadic Purkinje neurons throughout the anterior lobules (Figure 1C). These remaining anterior Purkinje neurons are eventually lost by nine weeks of age; at this time, additional Purkinje neuron degeneration is also observed throughout the central (lobules VI–VII) and posterior (lobules VIII–dorsal IX) zones (Figure 1D). This progressive spatiotemporal pattern of neurodegeneration eventually generates uniform Purkinje neuron death from lobule I/II to dorsal lobule IX in end-stage *Npc1^−/−^* mice, while the Purkinje neurons of the nodular zone (ventral lobule IX–X) are spared [29,31].

To identify metabolic or cellular pathways possibly contributing to the neuroprotection of the nodular zone Purkinje neurons in *Npc1^−/−^* mice, RNA-sequencing (RNA-seq) was performed on individual lobules previously demonstrated to be susceptible or resistant to Purkinje neuron degeneration. At 4.5 weeks of age, an age prior to the onset of widespread Purkinje neuron death, the cerebellum of *Npc1^+/+^* and *Npc1^−/−^* mice were microdissected and RNA was isolated from the early susceptible zebrin-II striped anterior lobule III, the later susceptible uniformly zebrin-II positive central lobule VI, and the uniformly zebrin-II positive resistant nodular lobule X. RNA-seq was then performed to compare gene expression patterns within the lobules of *Npc1^+/+^* and *Npc1^−/−^* mice and between the susceptible and resistant lobules of both genotypes. Each sample generated a minimum of 202 million reads per lobule. Between 84.7% to 89.5% of the reads were correctly mapped to the genome, with 47.9% to 65.9% of these mapped reads being properly paired reads.

### 2.2. Differential Gene Expression

Principal component analysis (PCA) showed that the samples clustered into two groups along the first component based on their localization to the anterior/central or nodular region of the cerebellum, with the first group comprised of lobule III and VI samples from both genotypes and the second group containing the lobule X samples corresponding to both genotypes. The large variability between anterior/central and nodular lobules, irrespective of genotype, was further visualized by volcano plots, in which the log_2_ fold change in gene expression and negative log_10_ adjusted *p*-value were plotted (Appendix A). The largest and most significant changes in gene expression were found in the comparison of anterior/central lobules to the nodular lobule within each genotype. For example, within the *Npc1^+/+^* cerebellum only 185 genes were differentially expressed when the anterior lobule III and central lobule VI were compared. In contrast, 1398 and 1313 genes were differentially expressed when either the anterior lobule III or the central lobule VI were compared to the nodular lobule X within the *Npc1^+/+^* cerebellum, respectively (Appendix A). Similarly, only 358 genes were differentially expressed within the anterior and central lobules of the *Npc1^−/−^* cerebellum, but 1317 and 1524 genes were differentially expressed when comparing either lobule III or lobule VI to lobule X, respectively (Appendix A). When comparing between the control and mutant cerebella, less numerous significant changes were observed, as comparing lobules III, VI, and X between genotypes only demonstrated 316, 79, and 94 differentially expressed genes, respectively (Appendix A). The complete differential expression analysis is provided for each comparison in Appendix A.

To further delineate the differences in gene expression that may contribute to the cerebellar pathology observed in *Npc1^−/−^* mice, we compared the significantly up/downregulated genes between the lobules of *Npc1^+/+^* and *Npc1^−/−^* mice (Appendix A). Sixty-six genes were commonly up/downregulated between lobules III, VI, and X of *Npc1^+/+^* and *Npc1^−/−^* mice, with six genes being downregulated and 60 genes being upregulated in all *Npc1^−/−^* lobules (Appendix A). Of the 60 genes that were significantly upregulated, multiple genes have been previously described as upregulated in the whole cerebellum of *Npc1^−/−^* mice (*Ctss*, *Cybb*, *Itgax*, *Itgb2*, *Gpnmb*, *Lyz2*, *Stab1*, *Icam1*, *C1qa*, *C1qb*) [35,36], the whole brain of *Npc1^−/−^* mice (*Clec7a*, *Mpeg1*, *Cd84*, *C4b*, *C3ar1*, *Trem2*) [37], used for biomarkers of NPC1 (*Lgals3* and *Ctsd*) [35], or as a marker for dysregulated microglia in *Npc1^−/−^* mice (*Cd22*) [38]. In almost all instances, the genes with upregulated expression in all lobules have been described as having a direct or indirect function related to the immune response. The most significantly downregulated gene in all lobules is *Npc1*, which validates our mouse model as being deficient in *Npc1* transcripts. The second most significantly downregulated gene was *Icmt*, which is a Purkinje neuron enriched gene that encodes for isoprenylcysteine carboxyl methyltransferase [39]. ICMT catalyzes the methylation of isoprenylated proteins that are targeted to the cell membrane. Dysregulation of *Icmt* has previously been shown to be an early marker of neurodegeneration in spinocerebellar ataxia type 1 [40]. Another gene downregulated in all *Npc1^−/−^* lobules is *Casq2*, which is another Purkinje neuron enriched gene that encodes for calsequestrin 2 [39]. Calsequestrin 2 has mostly been described in the context of cardiac function, where it acts as the main Ca^2+^ storage protein within sarcoplasmic reticulum [41]. As each of the most significantly downregulated genes, excluding *Npc1*, have been previously noted as being Purkinje neuron enriched, their downregulation across all lobules may represent early markers of neuronal dysfunction.

We also compared the gene expression changes that occurred specifically in the lobules of III and VI, but not X, between *Npc1^+/+^* and *Npc1^−/−^* mice (Appendix A). This comparison represents an interesting subset of 32 genes that are only differentially expressed in the regions of the cerebellum that are susceptible to neurodegeneration and thus may contribute to the spatiotemporal pattern of neurodegeneration observed in NPC1. Eleven genes were increased in expression exclusively in lobules III and VI of the *Npc1^−/−^* mice (Appendix A). The most significant increase in both lobules III and VI within *Npc1^−/−^* mice was for *Olfml3*, which is a microglia lineage marker [42]. Additional microglia activation markers, including *Lcp1* and *Ctsb*, were also included in the top five most significantly upregulated genes within lobules III and VI of the *Npc1^−/−^* cerebellum [43,44]. This enrichment of microglia-related genes specifically within the anterior lobules is reflective of the accumulation of microglia within this region of the cerebellum at an age prior to neurodegeneration [45].

Another gene significantly increased was tyrosine hydroxylase (*Th*), which enzymatically catalyzes the rate-limiting step in catecholamine biosynthesis. Previous reports have described ectopic expression of *Th* within the Purkinje neurons of various murine models of cerebellar ataxia, including *tottering*, *leaner*, *pogo*, *rolling Nagoya*, *dilute-lethal*, and *Npc1^−/−^* mice [31,46,47,48,49]. In these models, ectopic TH expression occurs within a subset of Purkinje neurons that form parasagittal stripes across the cerebellum that is analogous to zebrin II, but the expression of both proteins is not completely concordant [48,50,51]. Although the role of TH in the context of cerebellar ataxia has not been completely elucidated, *Th* expression is regulated by a calcium response element, and thus increased *Th* expression is believed to occur in response to the elevated intracellular calcium levels that occur in some cases of cerebellar ataxia [52,53,54]. Therefore, increased Th expression within lobules III and VI of *Npc1^−/−^* mice may reflect a dysregulation of intracellular calcium levels within this region of the cerebellum. This is of interest in NPC1, as an elevation of cytosolic calcium levels has been suggested to compensate for the reduced calcium concentration observed within the endolysosomal system of NPC1 cells and may improve the glycosphingolipid storage phenotype [5].

In addition to the genes with increased expression, 21 genes had decreased expression specifically within lobules III and VI of the *Npc1^−/−^* cerebellum (Appendix A). The most significantly downregulated gene was *Prkcg*, which is a calcium-activated serine/threonine-protein kinase that has been shown to be mutated in spinocerebellar ataxia type 14 [55]. Previous studies have also shown that *Prkcg* expression is enriched in Purkinje neurons and neuroprotective [56]; therefore, the decreased expression of *Prkcg* within the neurodegeneration-susceptible lobules III and VI of the *Npc1^−/−^* cerebellum could indicate that either this decreased expression predisposes Purkinje neurons within these regions to undergo cell death, or this decreased expression is occurring secondarily to the genetic insult and is only an early indicator of neuronal dysfunction.

We also performed an additional comparison for gene expression changes that occur only in lobule X of *Npc1^−/−^* mice, when compared to the same region of *Npc1^+/+^* mice (Appendix A). This comparison identified 40 genes that were differentially expressed exclusively in the region of the cerebellum that is spared from Purkinje neuron degeneration, and therefore these genes could confer a resistance to neurodegeneration to the cells of lobule X. Of the 40 genes differentially regulated in lobule X of *Npc1^−/−^* mice, 36 of these genes had increased expression in *Npc1^−/−^* mice (Appendix A). The most significantly elevated gene exclusively found within lobule X of *Npc1^−/−^* mice was *Thbs1*, which encodes for the thrombospondin 1 protein. THBS1 is a glia-derived factor that promotes the formation of synapses and that may be neuroprotective [57], and decreased *Thbs1* expression has been demonstrated as a marker of aging astrocytes within the cerebellum [58,59]. Therefore, it is possible that the increased expression of *Thbs1* within lobule X of *Npc1^−/−^* confers some neuroprotection within the lobule by improving astrocyte function. Although the above comparisons identified some novel gene expression changes that were localized to specific lobules within the *Npc1^−/−^* cerebellum, we sought additional methods to find *a priori* changes in expression that may help delineate the molecular mechanism leading to the spatiotemporal Purkinje neuron degeneration within NPC1.

### 2.3. Weighted Gene Co-expression Network Analysis

To identify gene expression patterns that may correlate with the anterior to posterior Purkinje neuron death observed in *Npc1^−/−^* mice, a Weighted Gene Co-expression Network Analysis (WGCNA) was performed on the RNA-seq data [34,60,61]. Fourteen different modules were detected through the WGCNA, and the eigengene for each module is depicted in Figure 2A. Using the eigengenes, the identified modules can be classified into two major groups based on gene expression among the *Npc1^+/+^* and *Npc1^−/−^* samples. The first group comprises eight different modules with NPC1-dependent (or disease-specific) expression patterns, in which the *Npc1^−/−^* expression is appreciably different from that of *Npc1^+/+^*. These eight modules were designated as NPC1-dependent-1–8. The second group, which contains six modules that have similar gene expression patterns when comparing *Npc1^+/+^* and *Npc1^−/−^* samples, were classified as having NPC1-independent expression patterns. These six modules were designated as NPC1-independent-1–6. Consistent with the principal component analysis (Figure 1E), the NPC1-independent expression modules contained the largest gene sets, as the NPC1-independent-1 module consisted of 3362 genes and the NPC1-independent-2 module consisted of 4448 genes (Figure 2B). These two large gene sets are distinguished by differential gene expression in lobule X relative to the anterior/central lobules, as the NPC1-independent-1 module is comprised of those genes with decreased expression in both *Npc1^+/+^* and *Npc1^−/−^* lobule X when compared to lobules III and VI, and the NPC1-independent-2 module consists of genes with increased expression in lobule X when compared to the lobules III and VI for both genotypes.

#### 2.3.1. NPC1-dependent Modules

To gain insight into the functional relevancy of the WGCNA modules, Ingenuity Pathway Analysis (IPA) was performed on the genes comprising each module. As a first analysis of the modules, networks comprising significant (*p*-value < 0.01) molecular and cellular functions were generated for each module. The functional network for the disease-specific NPC1-dependent-1 module (Figure 3A), whose eigengene demonstrates increased gene expression in all lobules of *Npc1^−/−^*, is highly interconnected and suggests that a common cell type or physiological response is responsible for the gene expression pattern of this module. This highly interconnected network results in a single functional cluster for the NPC1-dependent-1 module, which is represented by the most enriched term of cytotoxicity. All individual functions comprising this cluster appear to be directly or tangentially related to immune cell function, including leucocyte migration, proliferation of immune cells, and phagocytosis. Further pathway analysis showed a similar enrichment of pathways involved in the immune response, and the significantly enriched pathways for each module are provided in Appendix A.

The disease-specific modules of NPC1-dependent-2–5 (Figure 3B, Appendix A, respectively) show similar gene expression patterns to that of the NPC1-dependent-1 module, as all eigengenes showing elevated *Npc1^−/−^* gene expression across all lobules. All these clusters are related to cell movement, cell death, and immune activation. This demonstrates, in accordance in with previous publications [32,38,62], that the immune reaction appears to be homogenous throughout the cerebellum and therefore is not likely contributing to the pattered Purkinje neuron death observed in NPC1.

The remaining disease-specific modules, NPC1-dependent-6–8, have eigengenes characterized by decreased expression in the *Npc1^−/−^* samples across all lobules. The functional networks for both NPC1-dependent-6 (Appendix A) and NPC1-dependent-7 (Figure 3C) modules show sparsely populated and disconnected networks with little functional enrichment. The only clusters for the NPC1-dependent-6 module contain the enrichments of reassembly of organelle, metabolism of carbohydrate, ploidy of cells, and transport of protein. Likewise, the NPC1-dependent-7 module contained four enriched clusters, including mitotic catastrophe, DNA damage response, autophagy, and ploidy of cells. The functional network for genes belonging to the NPC1-dependent-8 module (Figure 3D), however, shows multiple large functional clusters, the most populated of which are morphology of Purkinje neurons, neurodegeneration, and metabolism of lipid. As these terms are all directly related to the pathology of NPC1, this module is particularly noteworthy.

The NPC1-dependent-8 module demonstrates a general decrease in anterior to posterior gene expression across the lobules for both genotypes, combined with highly decreased gene expression of the *Npc1^−/−^* sample in lobules III and VI but a negligible decrease in expression of the *Npc1^−/−^* sample within lobule X. Taken together, the NPC1-dependent-8 module is composed of genes with an expression pattern that differentiates between anterior and posterior regions and is also enriched for functional terms related to general neurodegeneration and more specifically, NPC1 (morphology of Purkinje neurons and metabolism of lipid). Therefore, the genes of the NPC1-dependent-8 module were subjected to further IPA analysis. In total, 21 canonical pathways were found to be significantly enriched in the NPC1-dependent-8 module (Appendix A). These include multiple pathways related to diacylglycerol and triacylglycerol biosynthesis (Appendix A). This *Npc1^−/−^* specific downregulation of genes related to di-and-triacylglycerol is in accordance with previously published reports that show triglycerides are reduced in both *Npc1^−/−^* mouse serum and hepatocytes [63,64]. Although the exact mechanism leading to the downregulation of triglycerides remains to be elucidated, it has been suggested that oleic acid or acetate-derived carbons, which normally are incorporated into triglycerides, are preferentially shunted into the generation of cholesterol [64]. The data gathered from these RNA-seq results would additionally suggest that the genes leading to the generation triglycerides are downregulated in *Npc1^−/−^*, potentially enabling the preferential metabolic flux of carbons into cholesterol synthesis at the expense of diacylglycerols and triacylglycerols.

The canonical pathway analysis of genes included in the NPC1-dependent-8 module also includes Amyloid Processing, which is of interest because amyloid-β (Aβ) release is increased and shifted towards the Aβ-plaque forming Aβ42 isoform in NPC1 patients [65]. Visualization of the pathway shows that the Aβ gene (*App*), along with the β-secretase genes (*Bace1* and *Bace2*), are included in the NPC1-dependent-8 module (Appendix A). Additional network analysis of the NPC1-dependent-8 module showed a highly significant gene network (score = 63) containing 34 genes all converging on *App* (Appendix A), suggesting that genes related to Aβ processing are altered in *Npc1^−/−^* and may contribute to the spatiotemporal neurodegeneration observed in the disease. More specifically, decreased β-secretase gene expression from anterior to posterior predicts possible decreased formation of Aβ-plaques in the posterior of the *Npc1^−/−^* cerebellum.

Previous studies have shown that β-secretase protein levels and activity are increased overall within the *Npc1^−/−^* cerebellum [66]. However, as the gene expression for *Bace1* and *Bace2* were included in the NPC1-dependent-8 module, we performed a β-secretase assay to determine if there was any lobule-specific alteration in β-secretase activity that could lead to decreased Aβ42 generation in the posterior lobules. Our results showed an increased activity in the anterior lobule of *Npc1^−/−^* samples, but this did not reach statistical significance when compared to the same lobules of the control mouse (Appendix A). The nodular lobule X, however, showed a significant increase in β-secretase activity (*p* = 0.023) in *Npc1^−/−^* samples compared to *Npc1^+/+^* samples. Additionally, no significant difference was observed between lobule III and lobule X of either genotype. These data are not reflective of the decreased *Bace1* and *Bace2* gene expression observed in lobule X of the *Npc1^−/−^*, as per our RNA-seq results, but rather may reflect a negative feedback loop due to the increased BACE1 protein levels previously reported in the *Npc1^−/−^* cerebellum [66].

#### 2.3.2. NPC1-independent Expression Modules

The WGCNA analysis also identified six modules where the cerebellar lobule gene expression was similar in both control and mutant tissues. We classified these as NPC1-independent expression modules (NPC1-independent-1 – 6). These NPC1-independent expression models had pronounced alterations in gene expression across the lobules, but negligible differences between genotypes. Similarly to for the disease-specific modules, networks were created to identify biologically relevant terms and to determine which modules would be investigated further.

The NPC1-independent expression modules of NPC1-independent-5 and 6 (Appendix A) do not contain gene expression patterns that differentiate between the anterior and posterior cerebellum. Furthermore, both modules are not significantly enriched in functional terms related to NPC1 (e.g., lipid metabolism, molecular transport, neurodegeneration, or neuronal morphology), and thus were not assessed further. In contrast to these modules, the NPC1-independent-3 and 4 modules (Figure 4A,B) display a significant and stepwise decrease or increase in gene expression from lobule III to X. Functional network analysis of the NPC1-independent-3 module shows the largest groupings being morphology of microvilli, metabolism of nucleotide, transport of cation, and transport of vesicles. Conversely, the NPC1-independent-4 module is comprised of functional groupings for cell-cell contact, metabolism of lipid, interphase of cells, and elongation of cells. Both NPC1-independent-3 and 4 modules contained enrichments of functional terms related to NPC1 (transport of vesicles and metabolism of lipid, respectively) and expression patterns that distinguish the anterior/central lobules to the nodular lobule. Therefore, these modules were subjected to further analysis.

As the NPC1-independent-3 and 4 modules demonstrated reciprocal expression patterns, the genes of both modules were combined for further pathway analysis. The utilization of genes with increased and decreased expression patterns across the lobules also enabled the use of a Z-score to determine the extent of activation (positive Z-score) or inhibition (negative Z-score) for each pathway, such that a positive Z-score indicates activation of that pathway in the posterior lobules and a negative Z-score indicates inhibition of the pathway in the posterior. In total, 40 pathways were significantly enriched in the combined NPC1-independent-3 and 4 modules, the top ten of which are shown in Appendix A and the significant pathways with the largest absolute Z-score are displayed in Appendix A. The Sonic Hedgehog (SHH) Signaling pathway (Appendix A), known to be critical for cerebellar patterning [67], is predicted to be increased in lobule X. Additionally, NPC1-specific abnormalities in SHH signaling have been described, leading to a reduction of ciliated cells in the *Npc1^−/−^* mice mouse brain [68]. During development, mRNA levels of *Shh* are decreased in *Npc1^−/−^* mice and likely results in defective proliferation of granule neurons within the cerebellum, as granule neuron proliferation is sustained by consistent levels of SHH [69]. Granule neuron function is important in the context of NPC1 and Purkinje neuron degeneration because they are the only excitatory neurons within the cerebellum and their parallel fibers within the molecular layer directly activate Purkinje neurons [70]. Therefore, the increase in SHH signaling as predicted by the pathway analysis of the NPC1-independent-3 and 4 modules may sustain granule neuron function and contribute to the survival of the Purkinje neurons within lobule X.

The functional network for the NPC1-independent-1 module, whose eigengene shows decreased expression specifically in lobule X of both *Npc1^+/+^* and *Npc1^−/−^*, has multiple distinct functional clusters that are comprised of individual functions with large gene sets (Figure 5A). The most populated clusters for the NPC1-independent-1 module are cell death, branching of neurons, movement of neurons, and morphology of synapse. Similar groupings of functions related to neuronal morphology and cell death are also seen in the NPC1-independent-2 module (Figure 5B). The NPC1-independent-2 module has the inverse expression pattern of the NPC1-independent-1 module, with elevated gene expression in lobule X of both genotypes. The largest functional clusterings for the NPC1-independent-2 module include cell death, branching of neurons, cell movement, and formation of intercellular junctions. Taken together, the combined functional clusterings of the NPC1-independent-1 and 2 modules demonstrate that the unique gene expression profile of lobule X leads to a significant enrichment for molecular and cellular function terms relating to neuronal morphology and cell death. Therefore, we further analyzed the genes from the NPC1-independent-1 and 2 modules by functional and pathway analysis to determine how they may contribute to the posterior cerebellum’s resistance to neurodegeneration.

As both modules are distinct in the gene expression of lobule X, the genes of the NPC1-independent-1 and 2 modules were combined for further pathway analysis to determine if the combination of genes act synergistically to contribute to the resistance to neurodegeneration seen in the posterior cerebellum. When combined, the genes of both modules generated 227 significantly enriched pathways. The top ten most significant pathways and those with the largest absolute Z-scores, with positive Z-scores indicating activation in lobule X and negative Z-scores indicating inhibition in lobule X, are displayed in Figure 6A,B. Included in the most significantly enriched pathways is that of Calcium Signaling. This pathway is of interest because calcium localization has been shown to be disrupted in NPC1, and furthermore, increases in cytosolic calcium levels have normalized the NPC1 cellular phenotype [5]. Examination of the Calcium Signaling pathway shows that numerous genes encoding both extracellular and intracellular calcium channels are altered in their gene expression pattern from the anterior to posterior cerebellum (Figure 6C). Although many genetic alterations are observed, almost all predict increased cytosolic concentration of calcium within lobule X. This suggests that increased cytosolic calcium levels may contribute to the cell survival in posterior cerebellum of NPC1, as calcium induces endosome-lysosome fusion and results in cellular protection in vitro [5,71].

Other pathways within the NPC1-independent-1 and 2 combined modules with large Z-scores include those for Dopamine Receptor Signaling (Appendix A) and Glutamate Receptor Signaling (Appendix A). The Z-scores for both pathways are both highly positive, with scores of 2.11 and 2.50, respectively, thus representing activation of both pathways in lobule X when compared to the anterior lobules. Within the Dopamine Receptor Signaling pathway, multiple subtypes of dopamine receptors demonstrate increased expression in lobule X, including *Drd2*, *Drd3*, *Drd4*, and *Drd5* (Appendix A). The expression of dopamine receptors has been shown to be decreased within the posterior lobules of patients with Parkinson disease, and the observed increase may reflect differential neurotransmitter signaling that may be neuroprotective in the posterior and nodular regions of the cerebellum [72]. Another modified neurotransmitter signaling pathway is that of Glutamate Receptor Signaling (Appendix A). As outlined in the pathway, many genes involved are expressed on glia cells and function to buffer extracellular glutamate, thus preventing excitotoxity from neighboring neurons [73]. This glutamate buffering is especially important at Purkinje neuron synapses as four out of the five glutamate transporters are localized to this region (GLAST/SLC1A3, EAAT2/SLC1A2, EAAT3/SLC1A1, and EAAT4/SLC1A6) [74,75,76]. Importantly, *Slc1a2* and *Slc1a6* are included in the NPC1-independent-2 module. Overall, the genes within the NPC1-independent-1 and 2 modules indicated in this pathway may lead to increased glutamate buffering in lobule X, which in turn could lead to both decreased neurotoxicity and neurodegeneration.

As the functional networks for both the NPC1-independent-1 and 2 modules show significant enrichment for functions related to cell death, further individual functional analysis was performed to determine if these functions coordinate to provide the neuroprotection observed in lobule X of the *Npc1^−/−^* mouse. All molecular and cellular functions classified into the categories of Cell Death or Cell Viability were isolated and plotted by Z-score (Figure 7A). All functions with a negative Z-score are related to cell death and specifically include the functions of degeneration of cells (−1.94), cell death of central nervous system cells (−1.479), and apoptosis of brain (−1.07). Additionally, all enriched functions related to cell survival had positive Z-scores, including cell survival (2.30), cell viability of tumor cell lines (2.13), and cell viability (1.88). Taken together, these results demonstrate a general inhibition of functions related to cell death in lobule X and an activation of functional terms related to cell survival in this same cerebellar region.

Although the previous results suggest that lobule X is more resistant to cell death, it is unclear if this is due to cell-autonomous survival functions within the Purkinje neurons of lobule X, or if the neuroprotection observed is due to other cellular factors within the lobule. The cell-autonomous neurodegeneration of Purkinje neurons has previously been demonstrated, but the factors leading to the spatiotemporal loss of these cells has not been determined [77,78]. To find individual genes that may be directly contributing to the neuroprotection of Purkinje neurons in lobule X, all NPC1-independent-1 and 2 module genes that were classified into functions of Cell Death or Cell Viability were identified. The average change in anterior to posterior gene expression via our RNA-seq data was then plotted against the Purkinje neuron gene enrichment previously reported by Doyle and colleagues [79] (Figure 7B). This plot specifically identifies: (1) genes that have been classified via functional analysis as pro-survival (green circles), pro-death (purple triangles), or context-dependent (orange diamonds) because they were classified as being pro-death and pro-survival in separate individual functions, (2) genes that have previously been identified as enriched in Purkinje neurons, with increasing Purkinje neuron specificity (displayed as a log_2_ fold change in the mRNA immunoprecipitated from Purkinje neurons (IP) versus its expression in the remaining tissue sample (UB) [79]), along the y-axis, (3) genes that have the largest anterior-to-posterior changes in gene expression (x-axis), with negative values representing higher gene expression in lobule III and VI and positive values representing increased gene expression in lobule X. When plotted together, there does not seem to be a Purkinje-neuron specific expression pattern that can account for the increased survival of Purkinje neurons within lobule X, as the Purkinje neuron specific genes with largest increases in expression in lobule X were noted as being pro-death (*Pla2g5*) or context-dependent (*Bub1* and *F2r*). Conversely, the Purkinje neuron specific genes with the highest anterior expression were noted as being pro-survival (*Fgf7*, *Cd28*, *Nek2,* and *Myof*). This result suggests that the Purkinje neuron neuroprotection observed within lobule X may not be cell autonomous.

The only exception to this is *Cck*, which is highly enriched in the Purkinje neurons of lobules III and VI and is classified as pro-death [80]. Cholecystokinin (CCK) is synthesized as a 115 amino acid prohormone that under post-translational processing to generate different biologically active CCK peptides. CCK33 is a 33 amino acid, gastrin-like peptide that was originally discovered in the gastrointestinal tract where it functions to increase gastric motility and induce the release of pancreatic enzymes [81]. Within the pancreas, CCK induces caspase activation, which leads to mitochondrial dysfunction and culminates in apoptosis [80]. In addition to its gastrointestinal functions, CCK is also one of the most abundant neuropeptides in the brain as the processed and sulfated form of CCK8 (CCK-8S) [82]. Within the brain, CCK-8S has been shown to modulate neuronal excitability by either directly acting as an excitatory neurotransmitter or by modulating the effects of other classical neurotransmitters [83,84]. Specifically, CCK-8S has been shown to suppress presynaptic GABA release in GABAergic interneurons and increase glutamate release from the hippocampus [85,86,87,88]. As CCK has been demonstrated to modulate both the apoptotic and neurotransmission pathways, and because of its anterior-enriched expression pattern in the cerebellum, CCK may contribute to the pathological cascade that predisposes the anterior lobules to early neurodegeneration.

### 2.4. Analysis of Purkinje Neuron Specific Transcripts

To determine if our RNA-seq data could identify lobule-specific changes in gene expression for Purkinje neuron enriched genes, we evaluated our data set against genes that have previously been identified using the Allen Brain Atlas as being both uniquely expressed in Purkinje neurons and differentially expressed between the anterior and nodular Purkinje neurons [89,90]. In accordance with previous studies, our RNA-seq data was able to detect differences in Purkinje neuron enriched genes that were previously described to have differences in anterior to posterior gene expression [89]. Examples of the gene expression patterns and corresponding in situ hybridization (ISH) images from the Allen Brain Atlas are included for genes with decreased expression in lobule X (Appendix A) and increased expression in lobule X (Appendix A). With this ability to accurately identify spatial differences in gene expression, we sought to identify additional novel Purkinje neuron specific genes that display a significant difference in lobule III/VI to lobule X cerebellar gene expression. Previous reports have identified genes that show enriched expression in Purkinje neurons [79] and additional studies have validated 38 of these as being Purkinje neuron specific [39]. From these 38 genes, we have identified 28 genes that are significantly up/downregulated in at least one comparison, either within the genotype comparisons or between the genotype comparisons. Specifically, we found 11 Purkinje neuron enriched genes that show no difference in gene expression when comparing between the lobules of *Npc1^+/+^* and *Npc1^−/−^* samples but do have significant changes in expression when comparing between lobules III, VI, and X (Appendix A). Four of the genes (*Slc1a6*, *Ebf1*, *Igfbp6*, and *Bean1*) were found to have increased expression in lobule X when compared to lobules III/VI in both *Npc1^+/+^* and *Npc1^−/−^* samples (Appendix A). This subset of genes includes *Slc1a6*, also known as EAAT4, which is one of the major glutamate transporters in the cerebellum and has previously been shown to be predominately expressed within the zebrin-II positive Purkinje neurons, which are preferentially found in the posterior lobules [76,91]. As described above, there are multiple changes in gene expression within the glutamate signaling pathway, and the highly significant increase in *Slc1a6* expression may protect the Purkinje neurons of lobule X from excitotoxicity in the context of NPC1. Additionally, this subset of Purkinje neuron enriched genes includes 7 genes (*Lmo7*, *Tgfb2*, *Homer3*, *Itpka*, *Dgkg*, *Kcnip1*, and *Cck*) that have significantly decreased gene expression in lobule X when compared to lobules III/VI of the same genotype (Appendix A).

In addition to those genes that showed no difference in expression between the genotypes, we also found 12 Purkinje neuron enriched genes that have differences in gene expression when comparing the same lobule between *Npc1^+/+^* and *Npc1^−/−^* samples (Appendix A). This includes two genes, *Casq2* and *Icmt*, which were found to be significantly decreased in lobules III, VI, and X of the *Npc1^−/−^* cerebellum (Appendix A). Two other genes (*Col18a1* and *Ryr1*) were found to be decreased in lobules III and VI of the *Npc1^−/−^* cerebellum (Appendix A). *Ryr1* functions as a calcium release channel and has been reported to play a pivotal role in maintaining dendritic elongation of Purkinje neurons [92]. Eight additional genes (*Ppp1r17*, *Calb1*, *Ptprr*, *B3gnt5*, *Ebf2*, *Rgs8*, *Atp2a3*, and *Doc2b*) were found to have decreased expression when comparing the anterior to posterior lobules of the *Npc1^+/+^* cerebellum but were also determined to have a significant decrease in gene expression between *Npc1^+/+^* and *Npc1^−/−^* samples within lobule III (Appendix A). These genes include *Calb1*, which is widely used as a marker of Purkinje neurons, and is of interest because immunofluorescence staining for CALB1 at this same time point (4.5 weeks of age) reveals no difference in Purkinje neuron density (Figure 1B). Therefore, this decrease may represent an early indicator of neurodegeneration that predates any disease-related pathological changes within the cerebellum.

In addition to the previous expression patterns, we also observed a set of seven Purkinje neuron enriched genes (*Trpc3*, *Lhx5*, *Cacna1g*, *Bcl11a*, *Grik1*, *Ptpn2*, and *Igfbp3*) that have no changes in gene expression across the lobules of the *Npc1^+/+^* samples but significantly decreased expression between *Npc1^+/+^* and *Npc1^−/−^* samples within lobule III (Appendix A). This set of genes included *Lhx5*, which is a transcriptional regulator that is required for Purkinje neuron development and differentiation [93]. Similar to the decrease in *Calb1* expression previously described, this *Npc1^−/−^* specific decrease of *Lhx5* within lobule III may be an earlier sign of neurodegeneration within the anterior cerebellum. Alternatively, these decreases in gene expression that were strictly restricted to lobule III *Npc1^−/−^*(observed in 14 out of 28 genes) could also predispose these cells to neurodegeneration.

To determine if these *Npc1^−/−^* specific decreases of Purkinje neuron gene expression within the anterior lobules represent the source of neurodegeneration, or merely the effect of impending degeneration, the larger set of Purkinje neuron enriched genes previously described by Doyle et al. were observed for their inclusion within our WGCNA analysis (Figure 8). This allowed us to surmise the gene expression pattern for each Purkinje neuron enriched gene based on the representative eigengene expression pattern for each module. After the modular inclusion was determined for each gene, an odds ratio was calculated of the relative number of Purkinje neuron enriched genes per module to the relative size of each module. The log_2_ of this odds ratio was then taken to evenly distribute the results around zero, with ratios greater than zero representing of an overrepresentation of Purkinje neuron enriched genes within the module. The modules with the highest overrepresentation of Purkinje neuron enriched genes were the NPC1-dependent-6, NPC1-dependent-7, and NPC1-dependent-8 modules. Interestingly, these three modules are the only ones in which the eigengene pattern depicts lower expression in *Npc1^−/−^* samples across the lobules (Figure 2A). In particular, the NPC1-dependent-6 eigengene shows a substantial decrease in *Npc1^−/−^* lobule III gene expression when compared to *Npc1^+/+^*, and only modest decreases in *Npc1^−/−^* gene expression between lobules III and VI when compared to those of *Npc1^+/+^*. These results are similar to those described within Appendix A, as an overrepresented sample of Purkinje neuron enriched genes demonstrate an expression pattern that is significantly decreased exclusively between *Npc1^−/−^* lobule III when compared to the same region of *Npc1^+/+^*. It is unlikely that the decreased expression of this large cohort of genes is required to predispose the anterior region of the cerebellum for neurodegeneration. However, it is more likely that this widespread decrease in Purkinje neuron enriched gene expression with the anterior cerebellum of *Npc1^−/−^* is an early indicator of damaged Purkinje neurons that will undergo neurodegeneration in the near future.

## 3. Discussion

In this study, RNA-sequencing was used to characterize spatial gene expression profiles of genes involved in patterned cerebellar degeneration observed in *Npc1^−/−^* mice. Specifically, the study sought to identify novel pathways associated with Purkinje neuron death, a highly affected cell type in NPC1. Based on our analysis, we compared gene expression changes between the lobules of *Npc1^+/+^* and *Npc1^−/−^* mice and changes within the lobules of each genotype. To our surprise, most of the significant difference in gene expression were observed between lobules III/VI and lobule X within each genotype (Appendix A). These observations were unexpected, as we anticipated finding significant changes in gene expression between *Npc1^+/+^* and *Npc1^−/−^* that could potentially explain the differences in neuronal survival observed in the NPC1 mouse model.

Genes differentially expressed between *Npc1^+/+^* and *Npc1^−/−^* samples revealed an overwhelming enrichment for genes related to the immune response and inflammation (see NPC1-dependent-1–4 modules of the WGCNA). Most of these genes show increased expression in *Npc1^−/−^* regardless of lobule, thus demonstrating that the immune response is ubiquitous throughout the cerebellum, in accordance with previous studies [36,94]. Although the immune response precedes any neurological symptoms (~2 weeks of age in mice) [32], modulation of the immune response by genetic deletion of both innate and adaptive immune related genes has shown little to modest effects on NPC1 disease progression [62,94,95,96]. These data, along with our work demonstrating ubiquitous cerebellar immune system activation, suggests that the inflammatory response is unlikely to underlie the patterned Purkinje neuron death observed in NPC1. Analysis of the other disease-specific WGCNA modules (NPC1-dependent-5–8) failed to identify any other pathways that could readily explain the patterned Purkinje neuron death observed in NPC1. These results suggest that the spatiotemporal pattern of Purkinje neuron degeneration is not the result of disease specific changes in the transcriptome.

Analysis of genes with different anterior/central lobule to nodular lobule expression patterns shows a multitude of potential reasons for neuroprotective nature of the Purkinje neurons within lobule X, including increased calcium signaling (Figure 6C), increased SHH signaling (Appendix A), and increased glutamate buffering (Appendix A). Most directly, functional analysis demonstrated a significant decrease in neuronal related cell death functions in lobule X and a concurrent increase in cell survival functions within this region (Figure 7A). As the Purkinje neurons are the most prominently affected cells in NPC1, we then sought to determine if there were genetic alterations intrinsic to Purkinje neurons that could explain the spatial differences in neurodegeneration; however, when restricted to Purkinje neuron enriched genes, there was only a single gene, *Cck*, that was classified as pro-death that was also enriched in expression in neurodegeneration susceptible lobules III and VI (Figure 7B). These results suggest that there are not cell-specific transcriptomic changes that are driving the neuroprotection of Purkinje neurons within lobule X, but rather that the overall environment of lobule X may drive the neuroprotection of Purkinje neurons within this specific region of the cerebellum.

This study has demonstrated that there are expansive and significant changes in gene expression between lobules III/VI and lobule X. The number of genes with significant changes in gene expression between anterior/central lobule compared to nodular lobule is over 10-fold higher than those comparisons done between the lobules of *Npc1^+/+^* and *Npc1^−/−^* mice (Appendix A). These large changes highlight the complexity of the cerebellum and the rigid regionality defined within it. Further studies are needed to determine whether the vast transcriptomic changes between the anterior/central and nodular cerebellar regions cause the difference in neurodegeneration observed between the regions.

To our knowledge, this is the first study to look at transcriptome-wide changes in gene expression between different cerebellar regions. To aid in the accessibility of this dataset, the gene expression results generated from this study are provided in interactive, web-based viewer at https://porterlab.shinyapps.io/cerebellarlobules/. Via this website, gene expression across the cerebellar lobules, and corresponding fold change comparisons, can be visualized for individual genes of interest.

## 4. Materials and Methods

### 4.1. Animals

All animal work conformed to the National Institutes of Health guidelines and was approved by the *Eunice Kennedy Shriver* National Institute of Child Health and Human Development Institutional Animal Care and Use Committee. Heterozygous *Npc1*^+/-^ mice (BALB/cNctr-*Npc1^m1N^*/J strain) were intercrossed to obtain control (*Npc1^+/+^*) and mutant (*Npc1^−/−^*) littermates. PCR genotyping was performed using tail DNA as previously described [97]. Mice were euthanized at 4.5 weeks of age by CO_2_ asphyxiation, transcardially perfused with phosphate buffered saline containing RNase Inhibitor (Thermo Fisher Scientific, Waltham, MA, USA), and the isolated cerebella were stored at −80 °C. The frozen cerebella of *Npc1^+/+^* and *Npc1^−/−^* mice were then microdissected on a bed of dry ice to isolate the anterior lobule III, the central lobule VI, as well as nodular lobule X. Lobules from three male and three female *Npc1^+/+^* or *Npc1^−/−^* mice were pooled together per each sample. A total of three pooled samples were collected for lobules III and VI, while four pooled samples were used for lobule X.

### 4.2. RNA-Sequencing

Total RNA was extracted from the lobule pools by homogenization in TRIzol reagent (Thermo Fisher Scientific), followed by further purification and concentration using RNeasy Mini Columns (Qiagen, Germantown, MD, USA). The quality and the quantity of RNA were assessed using both a Bioanalyzer (Agilent, Inc., Santa Clara, CA, USA) and NanoDrop (Thermo Fisher Scientific, Waltham, MA, USA). Total RNA (10 µg) was rRNA depleted in a twostep process first utilizing the Ambion RiboMinus Eukaryotes version 2 kit (Thermo Fisher Scientific, Waltham, MA, USA), as per manufacture instructions, followed by depletion using the Low Input RiboMinus Eukaryotes version 2 kit (Thermo Fisher Scientific, Waltham, MA, USA). rRNA-depleted RNA quality and concentration was verified after depletion by BioAnalyzer and Qubit (Thermo Fisher Scientific, Waltham, MA, USA) before proceeding to the library construction. Redundant cDNA library preparations were generated using the AB Library Builder System (Thermo Fisher Scientific, Waltham, MA, USA) on 300 ng of rRNA-depleted RNA, with a SOLiD total RNA-sequencing kit for paired-end reads (Thermo Fisher Scientific, Waltham, MA, USA). To reduce the potential for batch artifacts, the samples were randomly split into two pools, comprised of ten samples each. Following cleanup, the beads were deposited onto 12 lanes per pool and run on a SOLiD 5500 Sequencer (Thermo Fisher Scientific, Waltham, MA, USA) generating 75 bp by 50 bp paired-end reads. After sequencing, the color-space data was demultiplexed and aligned against the UCSC GRCm38/mm10 mouse genome with LifeScope using the paired-end whole transcriptome module (Thermo Fisher Scientific, Waltham, MA, USA).

### 4.3. Gene Expression Analyses

Raw read count tables were loaded into R (version 3.6.0) and differential expression analysis was conducted via DESeq2 (version 1.24.0), using the default parameters unless otherwise noted [98]. The raw reads were normalized and transformed using DESeq2′s regularized log (rlog) function, which was modeled with an experiment-wide trend of variance. The resulting rlog normalized counts were used for all subsequent depictions of gene count data. To test for differences in gene expression, pairwise contrasts were modeled for the comparisons of interest. Genes were considered significant if the adjusted *p*-values were less than 0.05, and as such, the independent filtering significance cutoff (alpha) was set to 0.05. To reduce the logarithmic fold changes for genes with high variability or low counts, the approximate posterior estimation for generalized linear model (apeglm) shrinkage estimator was utilized to obtain all log_2_-fold changes for each pairwise comparison [99]. Differential expression was defined by post-hoc filtering of adjusted *p*-value < 0.05 and an absolute log_2_ fold change >0.58 (corresponding to a fold change >1.5 in either direction). Principal component analysis was performed on normalized counts after surrogate variable analysis, using the sva package (version 3.28.0), was performed to remove unwanted variation from batch effects [100].

### 4.4. Weighted Gene Co-expression Network Analysis

The resulting counts tables were loaded into R (version 3.5.1). The Appendix A provide a detailed description of the methods used; here we briefly describe the workflow. First, genes with fewer than 10 reads in more than 15 samples were removed from analysis. Counts data were transformed using DESeq2′s rlog function. Surrogate variable analysis (sva, v3.28.0) was used to remove unwanted variation based on the study design and the detected surrogate variables were regressed out of the normalized count matrix. The normalized, SVA-corrected data were used in WGCNA analysis using the blockwiseModules function (see Appendix A for parameter details). For each gene, we report the kME (correlation of a gene with the eigengene of the module) across all modules, the *p*-value across all modules, the module assigned by WGCNA, and whether the WGCNA-assigned module had the best *p*-value or kME for that module.

### 4.5. Pathway Analysis

Pathway analysis was performed on the WGCNA modules using Ingenuity Pathway Analysis (IPA, https://www.qiagenbioinformatics.com/products/ingenuity-pathway-analysis). The analysis was performed using the default settings, and a *p*-value < 0.05 was considered statistically significant for all enriched pathways. Pathways were annotated using the molecular activity predictor within the IPA software.

Functional analysis was performed on all individual functions that are classified into one of the 33 categories that comprise the class of molecular and cellular functions, as defined within the IPA. Functional enrichment analysis of the molecular and cellular functions was performed with Cytoscape (version 3.6.0), using the EnrichmentMap app (version 3.1.0) [101]. Enrichment maps were generated with the number of nodes, representing individual functions, limited to a *p*-value cutoff of 0.01. The edges, representing the number of shared genes between each pair of functions, were set with the automatic setting and using the Overlap metric with a cutoff of 0.75. Functional clusters and labels were generated using the AutoAnnotate app (version 1.2) [102]. Clusters were annotated using the MCL Cluster algorithm, and labels were generated with the WordCloud: Adjacent Words algorithm, with the adjacent word bonus set at the default setting of 8. A maximum of four words were generated for each label, and these four words were edited down to create a coherent descriptor for each cluster.

### 4.6. Immunohistochemistry

Brain tissues for immunohistochemical analysis were processed as described previously [103]. Briefly, *Npc1^+/+^* and *Npc1^−/−^* mice at 4.5 weeks of were euthanized by CO_2_ asphyxiation and transcardially perfused with 4% paraformaldehyde in phosphate buffered saline (PBS). The brains were post-fixed for 24 h, then cryoprotected in 30% sucrose until the tissues sank. The brains were then sectioned parasagittally (20 µm) with a cryostat and the sections were collected in PBS supplemented with 0.1% Triton X-100 (PBSTx) (Sigma Aldrich, St. Louis, MO, USA). For calbindin staining, sections were blocked for one hour at room temperature with PBSTx supplemented with 10% goat serum (Sigma Aldrich, St. Louis, MO, USA). Sections were incubated overnight at 4 °C with primary antibody and detected with fluorescently labeled secondary antibody. All sections were counterstained using Hoescht 33342 (1:5000, Thermo Fisher Scientific), mounted and cover-slipped with Mowiol 4-88 mounting medium (Sigma Aldrich). The primary antibody used was anti-calbindin (1:2000, Cell Signaling Technologies, Danvers, MA, USA), with the corresponding secondary antibody of Alexa Fluor 594 goat anti-rabbit (1:5000, Thermo Fisher Scientific, Waltham, MA, USA).

### 4.7. β-secretase Assay

β-secretase activity was assayed using a β-Secretase Activity Fluorometric Assay Kit (Sigma Aldrich). Cerebellar lobule samples were isolated from *Npc1^+/+^* and *Npc1^−/−^* mice as described above, suspended in Extraction Buffer, and individual lobule lysates were prepared using a motor-driven pestle. For each lobule, four replicates of 10 µg lysate were loaded into a 96 well plate, and the assay was run according to the manufacturer’s instructions. The fluorometric results were obtained on a FLUOstar Omega (BMG Labtech, Ortenberg, Germany) with an excitation of 355 nm and an emission of 520 nm.

### 4.8. Statistical Analysis

Each data set was analyzed by two-way ANOVA and statistical significance was determined using Sidak’s multiple comparisons test between individual samples. The adjusted *p*-value, which is corrected for multiple comparisons, was utilized and a *p*-value of < 0.05 considered statistically significant. All statistics were calculated using GraphPad Prism software.

## Figures and Tables

**Figure 1 ijms-21-00292-f001:**
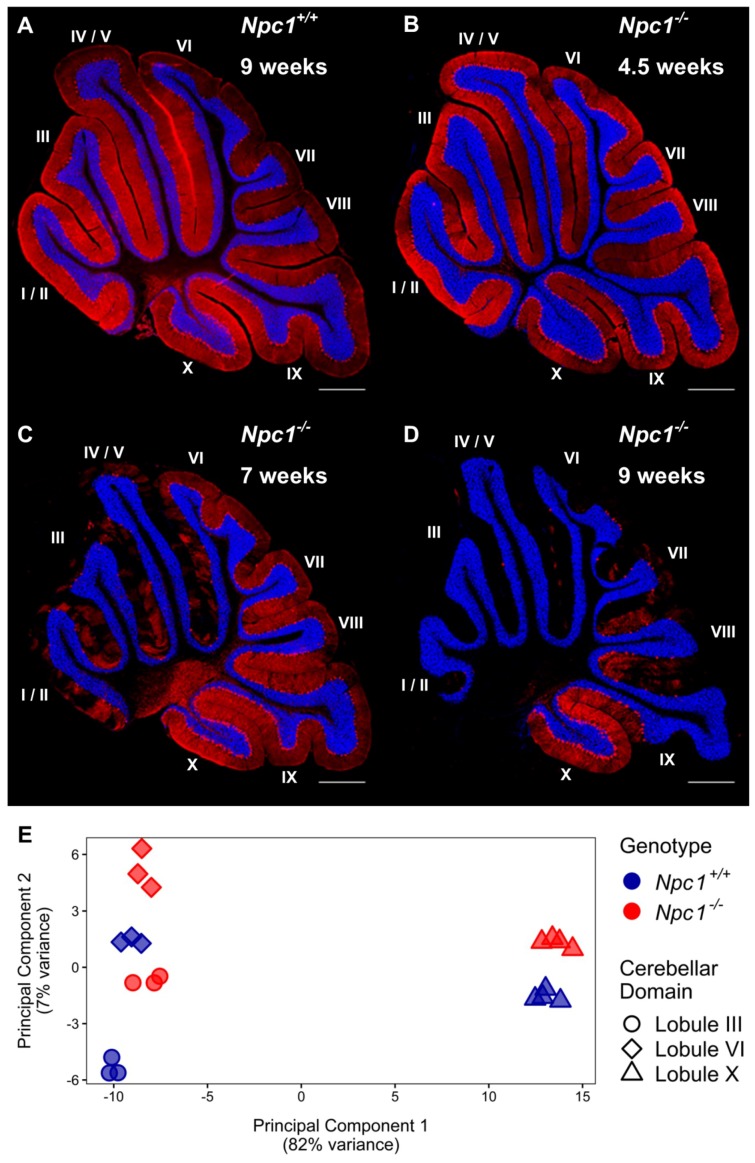
The transcriptome of cerebellar lobule X is distinct from other lobules, regardless of *Npc1* expression. (**A**–**D**) Calbindin 1 staining of Purkinje neurons in representative sagittal cerebellar sections of a *Npc1^+/+^* mouse at nine weeks of age (**A**), and *Npc1^−/−^* mice at 4.5 weeks (**B**), seven weeks (**C**), or nine weeks of age (**D**). The Purkinje neuron marker, Calbindin 1, is shown in red and a DAPI (4’,6-diamidino-2-phenylindole) counterstain in blue. Scale bar = 500 µm. (**E**) Principal component analysis of RNA-sequencing data sets from cerebellar lobules III, VI, and X from 4.5 weeks old *Npc1^+/+^* and *Npc1^−/−^* transgenic mice.

**Figure 2 ijms-21-00292-f002:**
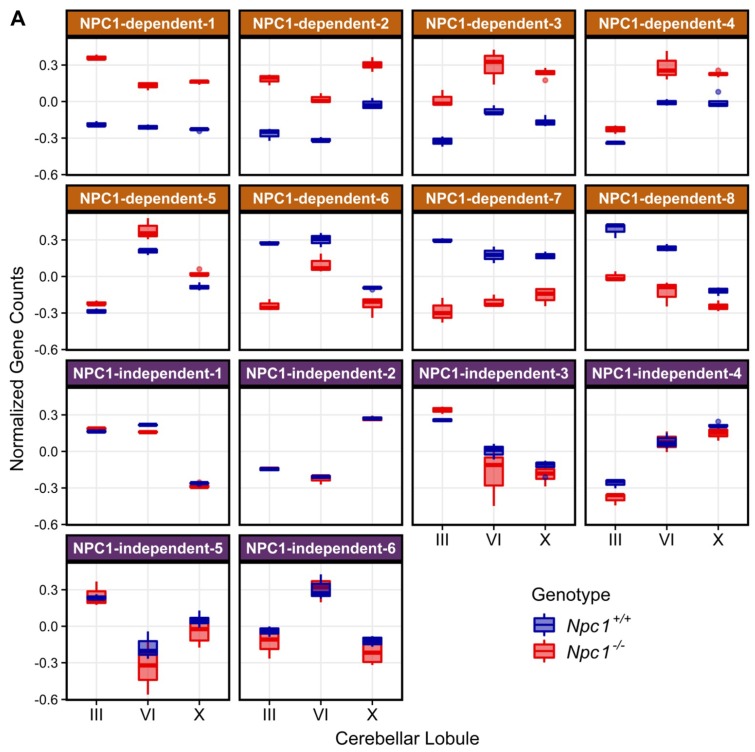
Weighted Gene Coexpression Network Analysis (WGCNA) of the RNA-sequencing data defined 14 distinct gene expression patterns in the *Npc1^+/+^* and *Npc1^−/−^* cerebellums from 4.5 weeks old mice. (**A**) The gene expression pattern, as depicted by an eigengene (a theoretical gene that describes the general pattern of the module), for each WGCNA module. The eight modules with a NPC1-dependent (or disease-specific) expression pattern, in which gene expression in different between the *Npc1^+/+^* and *Npc1^−/−^* samples, are depicted on the top half. The six modules with a NPC1-independent expression pattern, for which expression pattern changes across the lobules but not between the genotypes, are shown on the bottom. (**B**) The number of genes classified into each WGCNA module, with the percentage of genes in each module compared to the total in all modules shown in parenthesis.

**Figure 3 ijms-21-00292-f003:**
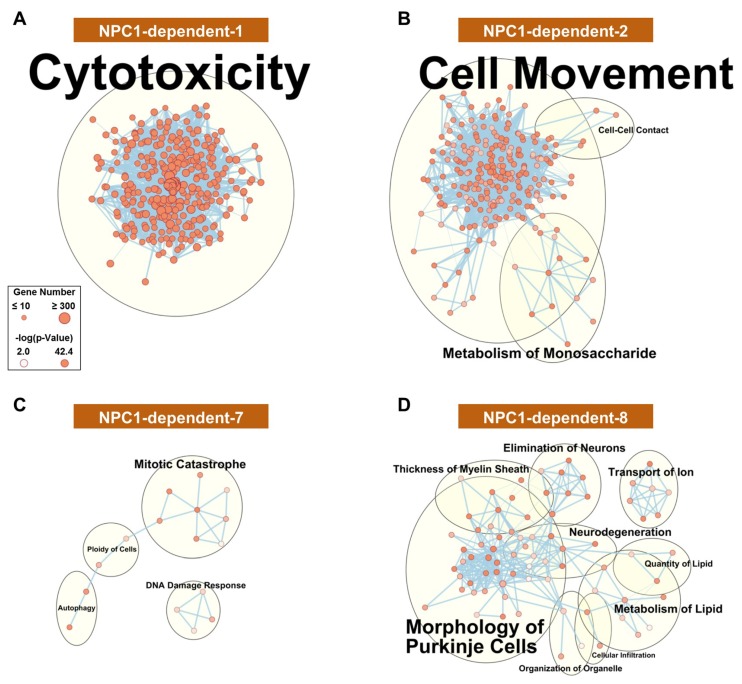
Functional enrichment for NPC1-dependent expression pattern modules. IPA functional analysis of the significantly enriched molecular and cellular functions for the selected disease-specific modules of NPC1-dependent-1 (**A**), NPC1-dependent-2 (**B**), NPC1-dependent-7 (**C**), and NPC1-dependent-8 (**D**). Each node represents a specific molecular and cellular function, the size of which corresponds to the number of genes included in each function. The thickness of the edges (lines connecting nodes) represent the number of shared genes between each function. Expression data is from 4.5 weeks old *Npc1* control and mutant mice.

**Figure 4 ijms-21-00292-f004:**
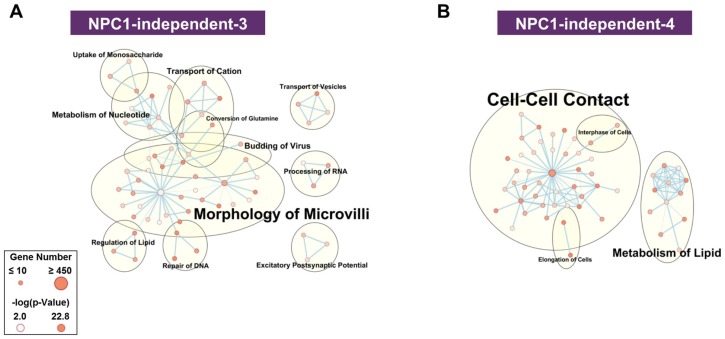
Functional enrichment for NPC1-independent-3 and 4 modules. IPA functional analysis of the significantly enriched molecular and cellular functions for the NPC1-independent-3 (**A**), and NPC1-independent-4 (**B**) modules. Each node represents a specific molecular and cellular function, the size of which corresponds to the number of genes included in each function. The edges represent the number of shared genes between each function. Expression data is from 4.5 weeks old *Npc1* control and mutant mice.

**Figure 5 ijms-21-00292-f005:**
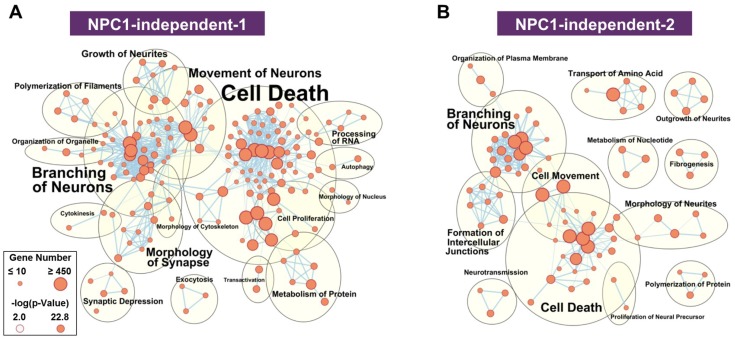
Functional enrichment for NPC1-independent-1 and 2 modules. IPA functional analysis of the significantly enriched molecular and cellular functions for the NPC1-independent-1 (**A**) and NPC1-independent-2 (**B**) modules. Each node represents a specific molecular and cellular function, the size of which corresponds to the number of genes included in each function. The edges represent the number of shared genes between each function. Expression data is from 4.5 weeks old *Npc1* control and mutant mice.

**Figure 6 ijms-21-00292-f006:**
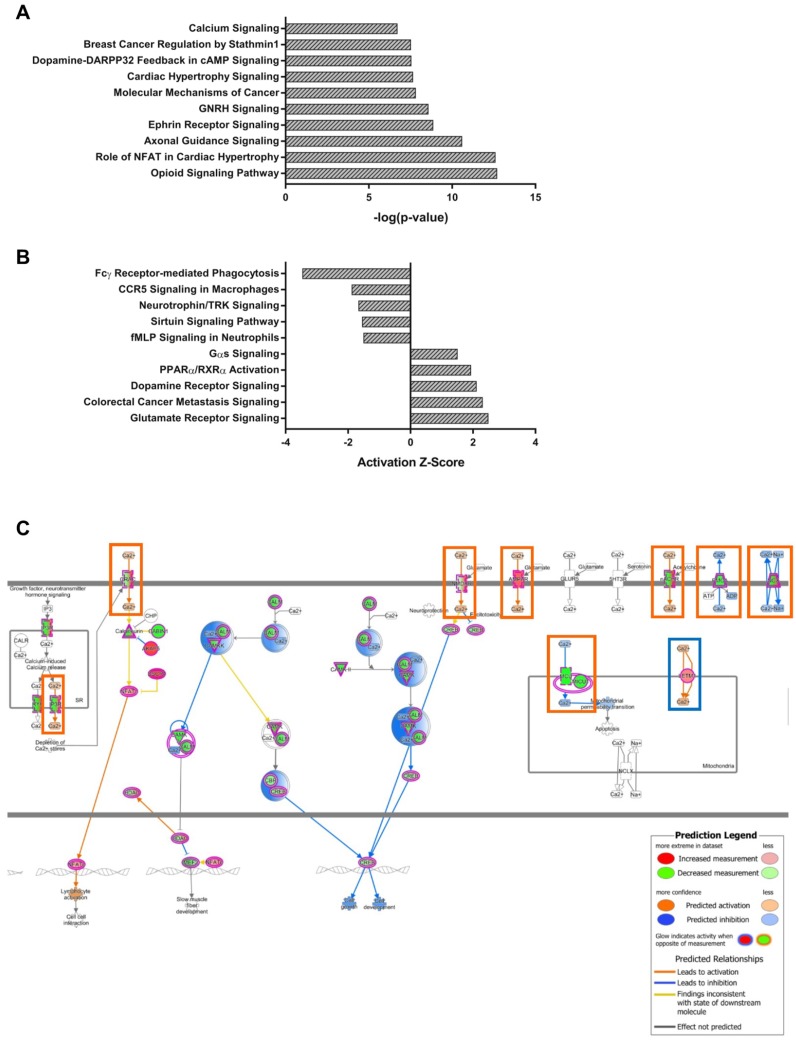
IPA canonical pathway analysis of the combined NPC1-independent-1 and 2 modules. (**A**) The top ten most significant IPA canonical pathways enriched within the gene sets of the combined NPC1-independent-1 and 2 modules. (**B**) The most significant IPA canonical pathways with Z-scores greater than 1.5 or less than -1.5. (**C**) The IPA canonical pathway for Calcium Signaling. Genes colored in red have increased expression in lobule X and those colored in green have decreased expression in lobule X, compared to the anterior lobules. The IPA generated molecular activity predictions are also depicted, with activation in lobule X depicted as orange lines and inhibition in lobule X depicted as blue lines. Gene expression predicting increased cytosolic calcium concentrations are highlighted in orange boxes, and gene expression predicting decreased cytosolic calcium are highlighted in blue boxes. Expression data is from 4.5 weeks old *Npc1* control and mutant mice.

**Figure 7 ijms-21-00292-f007:**
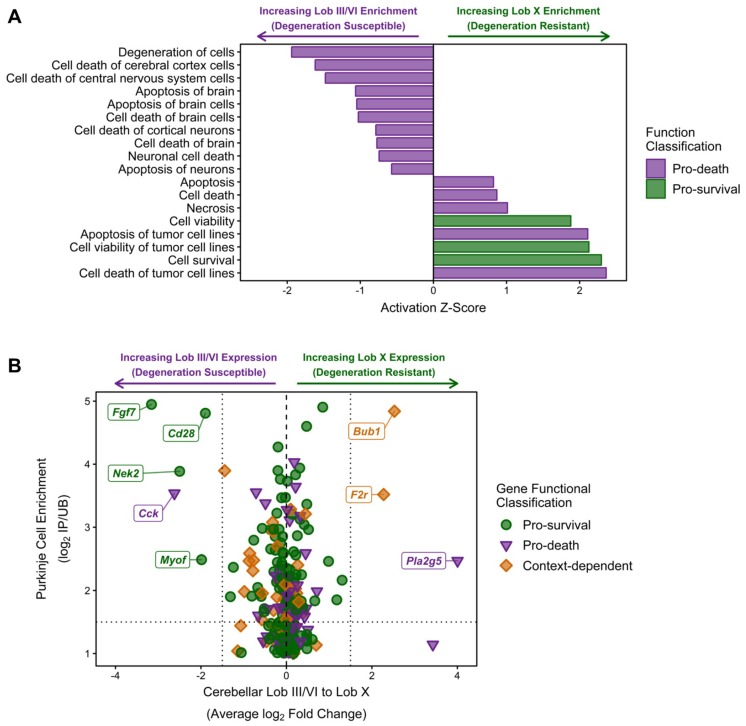
IPA analysis of cell death and survival functions in the combined NPC1-independent-1 and 2 modules. (**A**) IPA analysis of functions classified into the categories of cell death (purple bars) or cell survival (green bars). Negative Z-scores indicate inhibition in lobule X and Positive Z-scores indicate activation in lobule X. (**B**) Individual genes from the NPC1-independent-1 and 2 modules that were identified as increasing cell survival or increasing cell death. These genes were plotted against their average change in lobule III/VI to lobule X change in gene expression (log_2_ Fold Change) and the previously identified Purkinje neuron enrichment of gene expression (log_2_ IP/UB) [79]. The names for genes with a Purkinje neuron enrichment (log_2_ IP/UB > 1.5) and anterior or nodular specificity (log_2_ fold change < 1.5 for anterior or log_2_ fold change >1.5 for nodular) are noted. Expression data is from 4.5 weeks old *Npc1* control and mutant mice.

**Figure 8 ijms-21-00292-f008:**
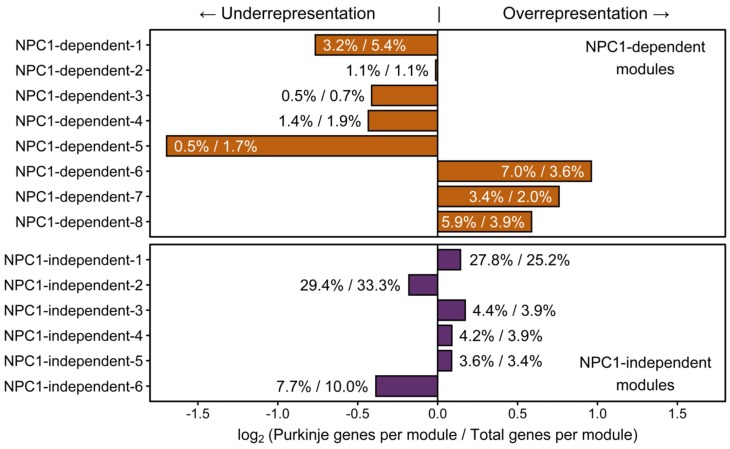
WGCNA module enrichment for Purkinje neuron enriched genes. For each WGCNA module, the percentage of Purkinje neuron enriched genes within that module compared to the total number of Purkinje neuron enriched genes is depicted as the numerator. The denominator is the percentage of genes within each WGCNA when compared to the total number of genes in all WGCNA modules. Modules with a log_2_ ratio of these two percentages greater than zero represent an overrepresentation of the number of Purkinje neuron enriched genes within that module, based on module size. Modules with a negative score have a paucity of Purkinje neuron enriched genes within that module, based on module size. Expression data is from 4.5 weeks old *Npc1* control and mutant mice.

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
