# Peer review of "Identification of Novel Pathways Associated with Patterned Cerebellar Purkinje Neuron Degeneration in Niemann-Pick Disease, Type C1"

_ijms, 2019, doi:10.3390/ijms21010292_

Round 1
Reviewer 1 Report
Martin and colleagues address a very important question, namely why only specific neurons degenerate (or degenerate faster than others) in NPC and other diseases. A prime example are Purkinje cells in the different lobes of the mouse cerebellum. To obtain new insight the authors compared transcript profiles across the different cerebellar lobes of wildtype and mutant mice using well-established methods including in-depth bioinformatic analyses. The manuscript is sound, the methods are appropriate, the results are highly interesting and well presented. This study represents a highly valuable contribution to the special issue. There are a few minor points that the authors should consider when revising the ms.
- Line 212: The statement "regulates the formation of astrocyte synapses" is odd and should be replaced. Thrombospondin is a glia-derived factor that promotes the formation of synapses and that may be neuroprotective: see Tran et al., 2019 PMID 31784286.
- Line 480-482: The statement "other cellular factors, independent of Purkinje neurons themselves" is rather nebulous and should be replaced by a more succinct conclusion. Moreover, this point raises the question whether the modules uncovered in this study can be mapped partially or entirely to specific cell types, namely microglia (Li et al., 2019 PMID: 30606613) or activated astrocytes (Liddelow et al., 2017 PMID: 28099414) using recently published transcript profiles.
- The authors should discuss/compare their data with respect to previous studies showing neuron-specific transcript changes in NPC1 (Demais et al., 2016 PMID: 27466344) and revealing factors that determine the neuronal vulnerability to pathologic changes (Tran et al., 2019 PMID 31784286).
Reviewer 2 Report
Dear Editor,
thank you for the opportunity to serve as reviewer for this very interesting manuscript. Here, the authors took advantage of RNA-seq to identify putative changes in target genes in the cerebellum of NPC1-/- and WT mice. When evaluated as a whole, this work further sustains the evidence that Purkinje neurons located at different cerebellar lobes are selectively susceptible to neurodegeneration. NPC1+/+ and NPC1-/- mice show an important and ubiquitous alteration in the expression of genes related to immune response and infammation in the cerebellum. Importantly, the authors demonstrated that gene expression differences in pro-death genes results to be highly pronounced independently on the genotype between lobules III/VI and X. Notably, posterior cerebellar lobes are enriched in genes involved in cell survival, further sustaining that posterior Purkinje cells are more resistant to neurodegeneration in NPC1.
The experimental design of this work is well conducted, results are interesting and properly integrated and discussed with the state of the art. Writing is good and renders this mauscript a pleasure to read.
I only have few minor concerns that should be addressed before considering this manuscript suitable for publication:
- Figure 4, 5 and 6 should be replaced with images at higher resolution as some details are not easily readable.
- In order to facilitate the comprehension of the readers, mouse age should be clearly indicated in every figure legend
Author Response
Reviewer 2
Comments and Suggestions for Authors
Dear Editor,
thank you for the opportunity to serve as reviewer for this very interesting manuscript. Here, the authors took advantage of RNA-seq to identify putative changes in target genes in the cerebellum of NPC1-/- and WT mice. When evaluated as a whole, this work further sustains the evidence that Purkinje neurons located at different cerebellar lobes are selectively susceptible to neurodegeneration. NPC1+/+ and NPC1-/- mice show an important and ubiquitous alteration in the expression of genes related to immune response and infammation in the cerebellum. Importantly, the authors demonstrated that gene expression differences in pro-death genes results to be highly pronounced independently on the genotype between lobules III/VI and X. Notably, posterior cerebellar lobes are enriched in genes involved in cell survival, further sustaining that posterior Purkinje cells are more resistant to neurodegeneration in NPC1.
The experimental design of this work is well conducted, results are interesting and properly integrated and discussed with the state of the art. Writing is good and renders this mauscript a pleasure to read.
I only have few minor concerns that should be addressed before considering this manuscript suitable for publication:
We appreciate these comments and address the minor concerns below.
Figure 4, 5 and 6 should be replaced with images at higher resolution as some details are not easily readable.
Figures were provided as high-resolution tiff files. We suspect the resolution issue arises when the files are embedded in the Word document. We will work with the publisher if this remains an issue.
In order to facilitate the comprehension of the readers, mouse age should be clearly indicated in every figure legend
This information was added to figure legends in the paper.
Reviewer 3 Report
In this study, Dr. Porter’s group examined Purkinje cell loss and RNA expression in Npc1 deficient mouse cerebellum. The research is well designed and conducted, and the manuscript and data (including additional figures and tables) will provide important information for the researchers in the study field.
Author Response
Reviewer 3
Comments and Suggestions for Authors
In this study, Dr. Porter’s group examined Purkinje cell loss and RNA expression in Npc1 deficient mouse cerebellum. The research is well designed and conducted, and the manuscript and data (including additional figures and tables) will provide important information for the researchers in the study field.
Thank you for this positive review.